# Indoor air pollution prevention practices and associated factors among household mothers in Olenchiti town, Oromia, Ethiopia

Worku Dugassa Girsha[ID]*, Alem Deksisa Abebe[☉], Ephrem Mannekulih Habtewold[☉], Meyrema Abdo Komicha[☉]

Department of Public Health, Adama Hospital Medical College, Adama, Ethiopia

☉ These authors contributed equally to this work.
* dugassaworku@gmail.com

## Abstract

### Introduction

Most households in low- and middle-income countries still cook using solid fuels in poorly ventilated dwellings. Indoor air pollution causes various health problems, like pneumonia, lung cancer, stillbirth, low birth weight, impaired cognitive development, and cataracts. Nevertheless, a few evidences are available in Africa, including Ethiopia. Therefore, this study aimed to assess the level of indoor air pollution prevention practices and associated factors among household mothers in Olenchiti town, Oromia, Ethiopia.

### Methods

A community-based cross-sectional study was conducted. Four hundred twenty mothers were randomly selected by systematic random sampling. Data was collected through an interview and observation checklist. The collected data entered into Epi-Info version 7.2.5 was cleaned, edited, and then exported to SPSS version 23 for analysis. Descriptive statistics were used to describe the findings. Binary logistic regression was computed to analyze the effect of each variable on the outcome variable. Model adequacy fitness was checked with the Hosmer-Lemeshow test. The multicollinearity of independent variables was checked with the variance inflation factor. Adjusted odds ratio with 95% confidence interval and $P$-value <0.05 was used as cutoff points to declare significance in the final model.

### Results

The overall good practices of mothers towards the prevention of indoor air pollution was 188 (45.0%). Mothers who had under-five children (AOR = 0.49, 95%CI (0.31–0.76), mothers in grade 9–12 (AOR = 0.51, 95%CI (0.28–0.92)) were significantly associated with indoor air pollution prevention practices.

### Conclusion

The overall good practices of mothers towards indoor air pollution were low compared to different findings. Under-five children and educational status were significantly associated with

**Data Availability Statement:** All relevant data are within the manuscript and its Supporting Information files.

**Funding:** Worku Dugassa Girsha wined the grant for the research. This study was entirely funded by the Oromia Regional Health Bureau. The total grant for the project was 156,000 Ethiopian birr (equivalent to 2,600 US$), website of Oromia Regional Health Bureau: Email: The funders had no role in study design, data collection and analysis, decision to publish, or preparation of the manuscript.ohbhead@ethionet.et.

**Competing interests:** The authors have declared that no competing interests exist.

indoor air pollution prevention practices in the final model. Therefore, the high school curriculums should include indoor air pollution topics.

## Introduction

Most households in low- and middle-income countries still cook using solid fuels (such as wood, crop wastes, charcoal, coal, and dung) and kerosene in poorly ventilated dwellings [1, 2]. Based on an analysis of household behavioral intervention practices of indoor air pollution (IAP), intervention should focus on the following behaviors: having separate kitchens, using modern stoves, keeping the room clean, using clean energy sources, keeping children at least two meters away from burning fires, open windows for ventilation while fires are burning, reduce the duration of solid fuel burning and avoid indoor cigarette smoking [3, 4].

Ensuring universal access to clean fuel and technologies is a target of the Sustainable Development Goal on Energy (SDG 7). However, Over half a billion people in Africa still rely on biomass fuel for cooking [5, 6]. About 90% of the population of Ethiopia uses biomass fuels to meet household energy needs [7]. Indoor air pollution causes high risks for women and children due to their domestic roles [8, 9]. Indoor air pollution causes various health problems, like pneumonia and acute lower respiratory infections [10, 11], chronic obstructive pulmonary disease (COPD) [12], lung cancer [13], stillbirth or low birth weight [14], impaired cognitive development [15], cataracts [16] and nasopharyngeal and laryngeal cancers [12].

There are research evidence gaps in prevention practices of indoor air pollution. Most of the research finding in developing countries including Ethiopia focused on measuring the pollution level of pollutants and related diseases. However, only a few studies are available on indoor air pollution prevention practices. The level of indoor air pollution prevention practices conducted in different African countries showed that, in the rural village of South Africa, almost half the households had good practices of indoor air pollution prevention methods [17], Cameroon, (77%) of households reported exclusive use of firewood [18], in Kenya thirty-seven percent of observed houses had no windows [19, 20], in Kasangati Town of Uganda, (11.5%) of respondents cooked inside their living houses [21].

In Ethiopia, different studies indicated that, in Northwest Ethiopia, (46%) [22] and rural Jimma zone, Ethiopia, (39%) [23] of households had good practices of indoor air pollution prevention methods. In Adama town, Ethiopia, (75.4%) of households used solid fuel for cooking in poorly ventilated houses [24]. Concerning factors affecting IAP, different studies conducted by the World Health Organization [25], in China [26], Bangladesh [27], rural Honduras [28], Kenya [19], Uganda [29], and Ethiopia [30, 31] revealed that income, education level, housing conditions, main fuel source and long hours burning solid fuels were significantly associated with the level of indoor air pollution practices.

Like other rural towns in Ethiopia, poorly ventilated housing, and limited access to separate cooking are visible features of living conditions in Olanchity town. Nevertheless, the present level of indoor air pollution prevention practices has not been studied in Olanchity town. Thus, the results of the study are used to create awareness for households, especially mothers to improve the efficient utilization of their household energy. It also helps the concerned governmental and nongovernmental organizations to design appropriate intervention strategies. Besides, it will be used by other interested researchers as a reference. Therefore, the study aims to assess the level of indoor air pollution prevention practices and associated factors among household mothers in the study area.

## Materials and methods

### Study area

The study was carried out in Olenchiti town, East Shoa zone, Orom5ia region, Ethiopia. The town is located 25 km from Adama town, along the road from Addis Ababa to Dire Dawa [32]. The town has two kebeles with a total population of 32,638 and 8,028 households. The town has one governmental hospital and health center.

### Study design and period

A community-based cross-sectional study design was conducted from the 1st of November 2022 to the 30th of April 2023.

### Source and study population

The source population includes all household mothers in Welenchiti Town. The study population was all household mothers in the selected kebele of Welenchiti town. Households with only a mother of age greater than or equal to 18 years were included in the study.

### Sample size determination and sampling procedures

The following assumptions were considered to determine the sample size: from a study done in Northwest Ethiopia, mothers had 46% indoor air pollution prevention practices [24]. Z-value at 95% confidence interval = 1.96, A 5% margin of error (d = 0.05). Additional 10% for non-response rate. Then sample size was calculated by using the formula estimate of single population proportion:

$$n = \frac{(Z_{\alpha/2})^2 * P(1-P)}{d^2} = \frac{(1.96)^2 * (0.46) * (0.54)}{(0.05)^2} = 382$$

Then adding an estimated 10% for the non-respondent rate the final sample size was 420. The town was stratified into kebeles to select a fairly representative sample of mothers from households. The sample size was distributed proportionally to each of the kebeles based on the number of household mothers they had. Kebele one had a total of 5803 household mothers (proportional allocation 304) and Kebele two had a total of 2225 household mothers (proportional allocation of 116). After assigning a number to each house and obtaining a sampling frame, each household mother sample was selected by systematic random sampling. The next household was substituted for households that didn't have the required mother. However, for households with more than one mother, the mother of the family representative was given priority, otherwise, selected by the lottery method.

### Study variables and data collection techniques

The dependent variable was indoor air pollution prevention practices and the independent variables included, socio-demographic factors (age, family size, under-five children, educational status, monthly income, marital status, occupation), housing-related factors (separate kitchen, presence of window, presence of chimney, type of stove, type of roof construction), household's behavioral factors (cigarette smoking, duration of cooking, complete combustion, opening window, knowledge on effects of indoor air pollution, training, supervision) and household's sources of energy (electricity, wood, charcoal, dung, kerosene).

An interviewer-administered semi-structured questionnaire [S1 File] adapted and modified from different literature was used to collect data with an observation checklist. The data were

collected by face-to-face interviews through house-to-house observation. Data collection was administered by four BSc in environmental health professionals and two supervisors who have a general master of public health. The household mothers were recruited from December 01/ 2022 to December 31/2022.

## Data quality assurance and measurements

A five percent pre-test of the instrument was done in Adama town. During the pre-test, a discussion was held with the interviewers on the problems they encountered in collecting data. The correction was incorporated in the final questionnaire. The training was given to both data collectors and supervisors a day before the pretest and a day after the pretest. The training includes the objectives of the study, method of data collection, checking completeness of the questionnaire, and the way to approach the households. Adjusted analysis was performed to account for potential confounders. Finally, Continuous supervision was done.

Indoor air pollution prevention practices were categorized into good practice or poor practice by considering the percentage of mothers who correctly answered or fulfilled the given six questionnaires such as the use of a separate kitchen, opening a window while cooking, use of a modern stove, outside complete combustion of charcoal, separating under five children while cooking, and use of clean energy sources.

## Data processing and analysis

The collected data were checked, entered into Epi-Info version 7.2.5, cleaned, edited, and then exported to SPSS version 23.0 for analysis [S1 Data]. Descriptive statistics were used to describe the findings. The normality of the age variable was checked with the Kolmogorov-Smirnov Test ($P<0.01$) and the data was not normally distributed. Binary logistic regression was computed to analyze the effect of each variable on the outcome variable. During bivariate logistic regression, independent variables with a P _value of $<0.25$ were considered candidates for the final logistic regression model [33]. Then all candidate variables were subjected to a multivariate logistic regression model to identify factors significantly associated with indoor air pollution practices after adjusting for possible confounding variables in the finally fitted regression model. Hosmer and Lemeshow test was performed to evaluate the model goodness of fit by comparing the disparities between observed and predicted values. The diagnosis results revealed that the model fit the data adequately (P-value = 0.44) and there was no significant difference between observed and expected values. The variance inflation factors (VIF) were used to assess if there were multicollinearity among covariates in the fitted model. The variance inflation factor greater than five was used as an indicator of multicollinearity [34].The strength of association was expressed in an adjusted odds ratio with 95% confidence interval and P -value $<0.05$ was used as cutoff points to declare the statistical significance in the final model.

## Ethical considerations

Ethical approval and clearance were obtained from the Adama Hospital Medical College, Institutional Ethical Review Committee. The health and administrative offices of the town received a letter of cooperation and ethical approval. Then, approval was obtained by contacting officials at various levels. The college institutional review board approved the information sheet and consent form, which was then attached to the questionnaire's first page [S1 File]. Before beginning to collect data, data collectors read each information sheet and consent form to enable mothers to make the right decisions. Furthermore, all the study participants were informed about the study's objective, benefits, and risks. Verbal consent of all study subjects was obtained before data collection. Participants were also informed that they have the full

right to discontinue or refuse to participate in the study. To ensure confidentiality, the name of the interviewee wasn't written on the questionnaire. Each respondent was assured that the information provided by them was confidential and used only for research. Moreover, no risk or harm was anticipated in the participation of the study.

## Results

### Socio-demographic characteristics of respondents

A total of 422 mothers were interviewed with a response rate of 99.52%. The median age was 37 years with an interquartile range of 16 years. The minimum and maximum age was 18 and 82 years respectively. One hundred forty-three (34.2%) of them were found in the age category of 31–40 years. One hundred eighty-five (44.3%) households had under five years children. Among the respondents, 284 (67.9%) were Oromo by ethnicity and 180 (43.1%) were Orthodox in religion. Regarding educational status, 105 (25.1%) had elementary (grade 1–8) levels and 192 (45.9%) were housewives in occupation. Regarding household monthly income, 179 (42.8%) got less than 36 US$ per month (Table 1).

### Housing conditions-related factors of households

Two hundred ninety-nine (71.5%) households had a separate kitchen from the main living room and 110 (26.3%) of household's kitchens or living rooms had windows and 122 (29.2%) of them had a chimney vent pipe for ventilation. One hundred fifty-four (36.8%) households used modern stoves for cooking and 291 (69.6%) of their roof were constructed with corrugated iron sheets (Table 2).

### Household's sources of energy

Among the studied households, Only 131 (31.3%) used electricity for cooking. However, most (99.3%) households used wood or charcoal for food preparation. Similarly, almost all (99%) households were using mixed sources of energy such as electricity, wood, charcoal, animal dung, and kerosene (Table 3).

### Household's behavioral factors

Only 19 (4.5%) of household members practiced indoor cigarette smoking. Two hundred ninety-two (69.9%) households opened their window or door while cooking. Around nine-tenths, (90.2%) of families did outside complete combustion of charcoal before entering the house. Only 69 (16.5%) households separated under five children while cooking (at least 2m from the burning fire). Most mothers (90.9%) continuously cooked in the kitchen for more than one hour without going outside the kitchen to get fresh air. Only a few mothers (2.6%) got training or follow-up on IAP from the health or energy bureau during the last six months. Forty-four households (10.5%) reported diseases during the last two weeks. Among the reported diseases, respiratory diseases accounted for 27 (61.4%) (Table 4).

### Knowledge of indoor air pollution

Most of the mothers, 386 (92.3%) and 406 (97.1%) mentioned indoor air pollution as a cause of respiratory diseases and cataracts or eye problems respectively. Some mothers also mentioned indoor air pollution as a cause of some health and related problems: lung cancer 262 (62.7%), low birth weight 220 (52.6%), impaired cognitive development 214 (51.2%), and global climate change 288 (68.9%). Concerning the overall knowledge of mothers, 279 (66.7%) had good knowledge of the impact of indoor air pollution (Table 5).

**Table 1. Socio-demographic characteristics among households in Olanchitiy town, East Shewa zone, Oromia, Ethiopia, 2023.**

| Variables (n = 418) | Number (%) |
|---|---|
| Family size | |
| < 5 | 284 (67.9) |
| ≥ 5 | 134 (32.1) |
| Having under-five children | |
| Yes | 185 (44.3) |
| No | 233 (55.7) |
| Age of respondents | |
| 18–30 | 133 (31.8) |
| 31–40 | 143 (34.2) |
| 41–50 | 89 (21.3) |
| 51 and above | 53 (12.7) |
| Ethnicity | |
| Oromo | 284 (67.9) |
| Amhara | 118 (28.2) |
| Gurage | 14 (3.3) |
| Others* | 2 (0.5) |
| Religion | |
| Muslim | 153 (36.6) |
| Orthodox | 180 (43.1) |
| Protestant | 80 (19.1) |
| Waqefata | 5 (1.2) |
| Educational status | |
| Unable to read and write | 104 (24.9) |
| Read and write | 71 (17.0) |
| Grade 1–8 | 105 (25.1) |
| Grade 9–12 | 91 (21.8) |
| Diploma and above | 47 (11.2) |
| **Occupation** | |
| Housewife | 192 (45.9) |
| Merchant | 94 (22.5) |
| Governmental employee | 33 (7.9) |
| Daily laborer | 31 (7.4) |
| Private employee | 39 (9.3) |
| Farmer | 29 (6.9) |
| Household monthly income (ETB) | |
| <2000 | 179 (42.8) |
| 2001–4000 | 161 (38.5) |
| 4001–5000 | 37 (8.9) |
| 5001 and above | 41 (9.8) |

* = Tigire and Kembata

## Household's indoor air pollution prevention practices

The overall good practices of households towards the prevention of indoor air pollution were 188 (45.0%) with 95%CI (40.2–51.0) (Table 6).

**Table 2. Housing conditions-related factors among households in Olanchitiy town, East Shewa zone, Oromia, Ethiopia, 2023.**

| Variables (n = 418) | Number (%) |
|---|---|
| Having a separate kitchen from the living room | |
| Yes | 299 (71.5) |
| No | 119 (28.5) |
| The kitchen or living room had a window | |
| Yes | 110 (26.3) |
| No | 308 (73.3) |
| The housing had a Chimney (vent pipe) | |
| Yes | 122 (29.2) |
| No | 296 (70.8) |
| Having a modern stove for cooking | |
| Yes | 154 (36.8) |
| No | 264 (63.2) |
| Type of modern stoves (n = 154) | |
| Mixadi* | 50 (32.5) |
| Electric stove | 73 (47.4) |
| Midija (lakech)* | 31 (20.1) |
| Type of roof construction | |
| Corrugated Iron sheet | 291 (69.6) |
| Traditional thatch roof | 125 (29.6) |
| Others (plastic) | 2 (0.5) |
| Room cleanliness (dust) | |
| Cleaned | 248 (59.3) |
| Not cleaned | 170 (40.7) |

* = Ethiopian locally-made modern stoves

**Table 3. Household's sources of energy for cooking in Olanchitiy town, East Shewa zone, Oromia, Ethiopia, 2023.**

| Variables (n = 418) | Number (%) |
|---|---|
| Electricity | |
| Yes | 131 (31.3) |
| No | 287 (68.7) |
| Wood | |
| Yes | 415 (99.3) |
| No | 3 (0.7) |
| Charcoal | |
| Yes | 415 (99.3) |
| No | 3 (0.7) |
| Dung | |
| Yes | 242 (57.9) |
| No | 176 (42.1) |
| Kerosene | |
| Yes | 35 (8.4) |
| No | 383 (91.6) |
| Mixed sources utilization | |
| Yes | 414 (99.0) |
| No | 4 (1.0) |

**Table 4. Household's behavioral factors on IAP in Olanchitiy town, East Shewa zone, Oromia, Ethiopia, 2023.**

| Variables (n = 418) | Number (%) |
|---|---|
| Indoor Cigarette smoking among households | |
| Yes | 19 (4.5) |
| No | 399 (95.5) |
| Opening window or door while cooking | |
| Yes | 292 (69.9) |
| No | 126 (30.1) |
| Outside Complete combustion of charcoal before entering the house | |
| Yes | 377 (90.2) |
| No | 41 (9.8) |
| Separation of under-five children while cooking | |
| Yes | 69 (16.5) |
| No | 349 (83.5) |
| Cooking in the kitchen for more than 1hrs without going out | |
| Yes | 380 (90.9) |
| No | 38 (9.1) |
| Training or follow-up was given on IAP by the health or energy bureau (6 months) | |
| Yes | 11 (2.6) |
| No | 407 (97.4) |
| Diseases reported among HH during the last two weeks | |
| Yes | 44 (10.5) |
| No | 418 (89.5) |
| Type of diseases reported | |
| Respiratory diseases | 27 (61.4) |
| Eye diseases | 13 (29.5) |
| Cancer | 4 (9.1) |

## Factors affecting household's indoor air pollution prevention practices

In the binary logistic regression variables such as family size (crude P–value <0.01), under-five children (crude P–value <0.01), educational status (crude P–value <0.01), occupation (crude P–value <0.11), household monthly income (crude P–value <0.01), and training (crude P–value <0.21), fulfilled the criteria and transferred to the multivariable analysis. However, after adjusting for possible confounding factors, only under-five children's and educational status was associated with indoor air pollution prevention practices at p-value ≤ 0.05. Concerning the variance inflation factors (VIF), the results of the diagnosis showed that all the covariate values of VIF were less than one point five (VIF-value < 1.5), which showed the absence of collinearity among covariates.

Among households who had under-five children 48% (AOR = 0.52, 95%CI (0.34–0.81) were less likely to practice indoor air pollution prevention practices than those who didn't have under-five children. Similarly, mothers in grade 9–12 educational level 48% (AOR = 0.52, 95%CI (0.28–0.98)) less likely to practice indoor air pollution prevention practices than those unable to read and write (Table 7).

## Discussions

The study assessed the level of indoor air pollution prevention practices and associated factors among households in Olenchiti town, East Shewa zone, Oromia, Ethiopia. Based on the main findings, less than half of the households practiced good indoor air pollution prevention,

**Table 5. Knowledge of IAP among mothers in Olanchitiy town, East Shewa zone, Oromia, Ethiopia, 2023.**

| Variables (n = 418) | Number (%) |
|---|---|
| IAP causes respiratory diseases | |
| Yes | 386 (92.3) |
| No | 32 (7.7) |
| IAP causes lung cancer | |
| Yes | 262 (62.7) |
| No | 156 (37.3) |
| IAP causes low birth weight | |
| Yes | 220 (52.6) |
| No | 198 (47.4) |
| IAP Impaired cognitive development | |
| Yes | 214 (51.2) |
| No | 204 (48.8) |
| IAP causes cataract | |
| Yes | 406 (97.1) |
| No | 12 (2.9) |
| IAP causes global climate change | |
| Yes | 288 (68.9) |
| No | 130 (31.1) |
| Overall knowledge | |
| Good Knowledge | 279 (66.7) |
| Poor knowledge | 139 (33.3) |

**Table 6. Household's IAP prevention practices in Olanchitiy town, East Shewa zone, Oromia, Ethiopia, 2023.**

| Variables (n = 418) | Number (%) |
|---|---|
| Use of separate kitchen | |
| Yes | 299 (71.5) |
| No | 119 (28.5) |
| Use of modern stove | |
| Yes | 154 (36.8) |
| No | 264 (63.2) |
| Opening window or door while cooking | |
| Yes | 292 (69.9) |
| No | 126 (30.1) |
| Separating children under five while cooking (at least 2m away from burning fires) | |
| Yes | 69 (16.5) |
| No | 349 (83.5) |
| Use of clean energy source (electricity) | |
| Yes | 131 (31.3) |
| No | 287 (68.7) |
| Outside complete combustion of charcoal | |
| Yes | 377 (90.2) |
| No | 41 (9.8) |
| Overall practice | |
| Good Practices | 188 (45.0) |
| Poor Practices | 230 (55.0) |

**Table 7. Factors affecting household IAP prevention practices in Olanchitiy town, East Shewa zone, Oromia, Ethiopia, 2023.**

| Variables | IAP prevention practices | | COR (95%CI) | AOR (95%CI) | P—value | VIF |
|---|---|---|---|---|---|---|
| | Good N (%) | Poor N (%) | | | | |
| Family size | | | | | | |
| < 5 | 111(39.1) | 173(60.9) | 2.10 (1.38–3.19) | 1.59(0.99–2.57) | 0.06 | 1.18 |
| ≥ 5 | 77(57.5) | 57(42.5) | 1.0 | 1.0 | | |
| Under five children | | | | | | |
| Yes | 106(57.3) | 79(42.7) | 0.40(0.27–0.60) | **0.52(0.34–0.81)** | **0.01** | 1.09 |
| No | 82(35.2) | 151(64.8) | 1.0 | 1.0 | | |
| Educational status | | | | | | |
| Illiterate | 36(34.6) | 68(65.4) | 1.0 | 1.0 | | 1.20 |
| Read and write | 27(38.0) | 44(62.0) | 0.86 (0.46–1.61) | 0.73 (0.37–1.44) | 0.37 | |
| Grade 1–8 | 43(41.0) | 62(59.0) | 0.76 (0.43–1.33) | 0.75 (0.42–1.37) | 0.36 | |
| Grade 9–12 | 48(52.7) | 43(47.3) | 0.47 (0.26–0.84) | **0.52 (0.28–0.98)** | **0.01** | |
| Diploma and above | 34(72.3) | 13(27.7) | 0.20 (0.09–0.43) | 0.37 (0.13–1.06) | 0.06 | |
| Occupation | | | | | | |
| Housewife | 82(42.7) | 110(57.3) | 0.51 (0.21–1.21) | 0.54(0.22–1.32) | 0.18 | 1.03 |
| Merchant | 50(53.2) | 44(46.8) | 0.33 (0.13–0.83) | 0.43(0.17–1.12) | 0.08 | |
| Governmental employee | 26(78.8) | 7(21.2) | 0.10 (0.03–1.32) | 0.31(0.07–1.30) | 0.11 | |
| Daily laborer | 7(22.6) | 24(77.4) | 1.30 (0.40–4.21) | 1.27(0.37–4.25) | 0.69 | |
| Private employee | 15(38.5) | 24(61.5) | 0.61 (0.21–1.72) | 0.94(0.31–2.84) | 0.92 | |
| Farmer | 8(27.6) | 21(72.4) | 1.0 | 1.0 | | |
| Household monthly income (ETB) | | | | | | |
| <2000 | 67(37.4) | 112(62.6) | 1.0 | 1.0 | | 1.22 |
| 2001–4000 | 74(46.0) | 87(54.0) | 0.70 (0.45–1.08) | 0.82 (0.51–1.32) | 0.42 | |
| 4001–5000 | 17(45.9) | 20(54.1) | 0.70 (0.34–1.43) | 1.04 (0.47–2.32) | 0.90 | |
| 5001 and above | 30(73.2) | 11(26.8) | 0.21 (0.10–0.46) | 0.49 (0.20–1.18) | 0.11 | |
| Training | | | | | | |
| Yes | 7(63.6) | 4(36.4) | 0.45 (0.13–1.58) | 0.57(0.14–2.25) | 0.43 | 1.03 |
| No | 181(44.5) | 226(55.5) | 1.0 | 1.0 | | |

Abbreviations: COR crude odds ratio, AOR adjusted odds ratio, CI confidence interval, 1.0 reference category, VIF variance inflation factors

(45.0%, 95%CI (40.2–51.0)). The finding was similar to a study done in a rural village in South Africa, 51% [19] and in Northwest Ethiopia, (46%) [24], however, it was higher than the study done in rural Jimma, (39%) [25], the difference may be due to the study time variations and methods.

Regarding household energy sources, most (68.7%) households used biomass fuel for cooking. The finding was almost similar to a World Bank report, in Africa, over half a billion people still rely on biomass fuel for cooking and heating [6]. This finding was much behind the target of the Sustainable Development Goal on Energy (SDG 7) which says to ensure universal access to clean fuel and technologies.

Concerning the housing condition of households, (28.5%) households had no separate kitchen from the main living room and cooked inside their living houses. The finding was similar to a study done in a rural town in Kenya (37%) [22], however, the finding was higher than the study done in a study conducted in rural Town of Uganda (11.5%) [23], with a slight difference may be due to study time variation and methods. Similarly, only (36.8%) of households used improved or modern stoves for cooking, similar to a study done in a rural village in

South Africa (27%) [19]. Regarding household behavioral factors, (69.9%) of households opened windows or doors while cooking. Similar to a study done in a rural village in South Africa (69%) [19], the possible reasons may be the knowledge of mothers regarding the impact of indoor air pollution on the household members should be increased.

Households that had under-five children were 51% less likely to practice indoor air pollution prevention practices than those who didn't have under-five children. Many findings suggest that mothers with under five children are so busy, that much of their time is wasted on child caring, cooking, family guidance, and income generation. Similarly, mothers in grades 9–12 educational level (49%) were less likely to practice indoor air pollution prevention practices than those unable to read and write. The finding was contrary to studies done in China [28] and Bangladesh [19] which suggested that households with highly educated adults practice IAP prevention more likely than the least-educated households. The difference may be in the study area, about IAP topics not included in their high school curriculum and attention was not given to high school students.

## Limitations of the study

The study findings included household mothers from a single town. Therefore, the research results are limited to this particular town and cannot be generalized to other towns in Ethiopia.

## Conclusions and recommendations

The overall good practices of households towards the prevention of indoor air pollution were low compared to different findings in Ethiopia as well as in Africa. Under-five children and educational status were significantly associated with indoor air pollution prevention practices in the final model. Therefore, this study recommends the following actions to alleviate indoor air pollution problems in Welenchiti town. The town health office should strengthen family planning services for households to regulate their family size. Similarly, the town educational offices should incorporate indoor air pollution topics into regular high school curriculums. Further research is needed for measuring the concentration level of indoor air pollutants.

## Supporting information

**S1 File. Information sheet, consent form and English version questionnaires.**
(PDF)

**S1 Data. SPSS data set of study participants.**
(SAV)

## Acknowledgments

We hugely acknowledge the different levels of administrative hierarchy. Furthermore, the authors like to acknowledge the study participants, data collectors, and colleagues for their genuine cooperation during the study.

## Author Contributions

**Conceptualization:** Worku Dugassa Girsha, Alem Deksisa Abebe, Ephrem Mannekulih Habtewold, Meyrema Abdo Komicha.

**Data curation:** Worku Dugassa Girsha, Alem Deksisa Abebe, Ephrem Mannekulih Habtewold, Meyrema Abdo Komicha.

**Formal analysis:** Worku Dugassa Girsha, Alem Deksisa Abebe, Ephrem Mannekulih Habtewold, Meyrema Abdo Komicha.

**Funding acquisition:** Worku Dugassa Girsha, Alem Deksisa Abebe, Ephrem Mannekulih Habtewold, Meyrema Abdo Komicha.

**Investigation:** Worku Dugassa Girsha, Alem Deksisa Abebe, Ephrem Mannekulih Habtewold, Meyrema Abdo Komicha.

**Methodology:** Worku Dugassa Girsha, Alem Deksisa Abebe, Ephrem Mannekulih Habtewold, Meyrema Abdo Komicha.

**Project administration:** Worku Dugassa Girsha, Alem Deksisa Abebe, Ephrem Mannekulih Habtewold, Meyrema Abdo Komicha.

**Resources:** Worku Dugassa Girsha, Alem Deksisa Abebe, Ephrem Mannekulih Habtewold, Meyrema Abdo Komicha.

**Software:** Worku Dugassa Girsha, Alem Deksisa Abebe, Ephrem Mannekulih Habtewold, Meyrema Abdo Komicha.

**Supervision:** Worku Dugassa Girsha, Alem Deksisa Abebe, Ephrem Mannekulih Habtewold, Meyrema Abdo Komicha.

**Validation:** Worku Dugassa Girsha, Alem Deksisa Abebe, Ephrem Mannekulih Habtewold, Meyrema Abdo Komicha.

**Visualization:** Worku Dugassa Girsha, Alem Deksisa Abebe, Ephrem Mannekulih Habtewold, Meyrema Abdo Komicha.

**Writing – original draft:** Worku Dugassa Girsha, Alem Deksisa Abebe, Ephrem Mannekulih Habtewold, Meyrema Abdo Komicha.

**Writing – review & editing:** Worku Dugassa Girsha, Alem Deksisa Abebe, Ephrem Mannekulih Habtewold, Meyrema Abdo Komicha.

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
