## [Editor Report · Decision Letter 0]

23 Jul 2023

PONE-D-23-21169Indoor air pollution prevention practices and associated factors among household mothers in Olenchiti town, Oromia, EthiopiaPLOS ONE

Dear Dr. Girsha,

Thank you for submitting your manuscript to PLOS ONE. After careful consideration, we feel that it has merit but does not fully meet PLOS ONE’s publication criteria as it currently stands. Therefore, we invite you to submit a revised version of the manuscript that addresses the points raised during the review process.

We look forward to receiving your revised manuscript.

Kind regards,

Srijan Lal Shrestha, Ph.D.

Academic Editor

PLOS ONE

- https://article.sciencepublishinggroup.com/html/10.11648.j.sjph.20160406.13.html

In your revision ensure you cite all your sources (including your own works), and quote or rephrase any duplicated text outside the methods section. Further consideration is dependent on these concerns being addressed."""

3. In the ethics statement in the Methods, you have specified that verbal consent was obtained. Please provide additional details regarding how this consent was documented and witnessed, and state whether this was approved by the IRB.

“Worku Dugassa Girsha wined the grant for the research. This study was entirely funded by the Oromia Regional Health Bureau. The total grant for the project was 156,000 Ethiopian birr (equivalent to 2,600 US$), website of Oromia Regional Health Bureau: Email: The funders had no role in study design, data collection and analysis, decision to publish, or preparation of the manuscript.ohbhead@ethionet.et.”

Additional Editor Comments:

The paper is about indoor air pollution prevention practices and its associated factors based upon cross-sectional study design. The area of research is interesting and suitable for publication in PLOS ONE. The editorial review of the paper found the following.

1. Use of sampling design particularly systematic sampling has not been explained properly. Usually, systematic sampling is implemented from the available sampling frame which is not mentioned in the manuscript. If done without listing of households, how exactly systematic sampling was implemented has not been clearly mentioned. Stratification statistics needs to be mentioned as well.

2. Logistic regression results are not clearly reported. It has been mentioned independent variables are included if p value < 0.25. However, reporting suggests otherwise. There is absence of model adequacy tests for the model. No goodness of fit tests reported. The reference categories are not appropriately selected so as to reflect the computation of risk from odds ratios.

3. Data availability statement provided by authors is not up to the mark. As per the statement "data can be made available upon request" is not normally acceptable.

4. Authors individual contributions in the paper are not clearly mentioned.

Upon considering above, the paper is not suitable to be sent for peer review in the current context. If the authors revise the paper strictly as per the editorial comments and re-submit the paper with acceptable data availability statement and clarifications of authors contributions specifically mentioned, then it can be considered for editorial review again.

Academic editor

---

## [Author Response · Author response to Decision Letter 0]

12 Sep 2023

First of all, thank you all, for your nice comments and suggestions, we try to correct the comments as the following: 

1. We tried to edit our manuscript according to PLOS ONE's style requirements 

2. Rephrased and cited overlapping text with the previous publication

3. Ethical statements are clarified in the document 

4. The funding is more clarified as below:

Worku Dugassa Girsha wined the grant for the research. This study was entirely funded by the Oromia Regional Health Bureau. The total grant for the project was 156,000 Ethiopian birr (equivalent to 2,600 US$). The Oromia Regional Health Bureau funded only the indicated financial support, whereas, our institution, Adama hospital medical college provided vehicles with fuel for supervision. All authors are permanent workers of Adama hospital medical college and received regular monthly salaries, but not related to this study. Website of Oromia Regional Health Bureau: Email: ohbhead@ethionet.et. The funders had no role in study design, data collection, and analysis, decision to publish, or preparation of the manuscript. 

5. The following Supporting information is provided 

a. S1 File. (Information sheet, consent form, and English version questionnaires (PDF)

b. S1 Data (SPSS Data Set) (SAV)

6. For systematic sampling, sampling frame or list and Stratification statistics were clarified 

7. For logistic regression, basic assumptions (normality, model fitness, and multicollinearity) were checked and revised (included in the manuscript)

8. Authors' contributions corrected according to guideline contents

9. A marked-up copy of our manuscript that highlighted changes with yellow color and an unmarked version of our revised paper without color changes are attached. 

10. Laboratory protocols are not used or applied in the study and we are free of that.

With all the best regards 

Worku Dugassa Girsha

---

## [Decision Letter · Decision Letter 1]

12 Oct 2023

PONE-D-23-21169R1Indoor air pollution prevention practices and associated factors among household mothers in Olenchiti town, Oromia, EthiopiaPLOS ONE

Dear Dr. Girsha,

Thank you for submitting your manuscript to PLOS ONE. After careful consideration, we feel that it has merit but does not fully meet PLOS ONE’s publication criteria as it currently stands. Therefore, we invite you to submit a revised version of the manuscript that addresses the points raised during the review process.

We have received peer review comments on your manuscript. The reviewers have suggested to revise the paper as per the major and minor comments made. Please go through the comments thoroughly and revise the paper accordingly.

We look forward to receiving your revised manuscript.

Kind regards,

Srijan Lal Shrestha, Ph.D.

Academic Editor

PLOS ONE

Additional Editor Comments:

Dear Author Girsha, MPH

Peer review comments of your submission have been received. Reviewers have suggested to revise your manuscript as per the comments made. Please go through the comments and address them appropriately and submit the revised manuscript.

Reviewers' comments:

Reviewer's Responses to Questions

**Comments to the Author**

1. If the authors have adequately addressed your comments raised in a previous round of review and you feel that this manuscript is now acceptable for publication, you may indicate that here to bypass the “Comments to the Author” section, enter your conflict of interest statement in the “Confidential to Editor” section, and submit your "Accept" recommendation.

Reviewer #1: (No Response)

Reviewer #2: All comments have been addressed

2. Is the manuscript technically sound, and do the data support the conclusions?

Reviewer #1: Partly

Reviewer #2: Yes

3. Has the statistical analysis been performed appropriately and rigorously? 

Reviewer #1: No

Reviewer #2: (No Response)

4. Have the authors made all data underlying the findings in their manuscript fully available?

Reviewer #1: Yes

Reviewer #2: Yes

5. Is the manuscript presented in an intelligible fashion and written in standard English?

Reviewer #1: Yes

Reviewer #2: Yes

6. Review Comments to the Author

Reviewer #1: This a very nice and sound study but there are rooms for improvement in technical side.

Line 115-16: Since the sample size is large in both the strata (>100) test of normality must be done with the Kolmogoro-Smirnov test of normality and same test result must be reported. Shapiro-Wilk test is used for small samples only. SPSS reports this test with Lilliefores correction.

Line 117-18: Provide reference for this higher p-value cut-off of 0.25. Do not write this as statistically significant in the result, it only suggests possible significance in the final model after controlling for other variables.

Line 117-19: You must use the sampling weight while doing bi-variate and multi-variate analysis (Table 1 to Table 7). Make sure to use design weight obtained from the random selection using sampling frame and non-response correction weight while calculating the sampling weight. Kindly use stratification variable too in the weight calculation. Prepare a separate Excel file for the same and upload it as supplement file and add that final sampling weight values in the SPSS data as well and fit the bi-variate and multi-variate logistic regression using the computed sampling weight.

Line 119: Write "multivariate logistic regression model" instead of "multiple regression analysis". Multiple is used for linear regression model only.

Line 123-24: VIF values must be reported as separate column in Table 7 after p-value for each variable. Kindly mention the VIF cut-off value used to rule out the presence of multicollinearity in the logistic regression with reference.

Reviewer #2: The following are the suggestion and feedback to the author

Line numbers

15- for which variable did you test the normality and why it was necessary here?

23- towards prevention from

80- Z value at 95% confidence interval =1.96

87/88- what was the value of sampling interval, it is possible to that large population ?

121- What is the value of Hosmer Lemeshow value?

123- what is the value of VIF?

142- does knowledge vary according to such huge range of age Like 18 to 82 yrd?Table 149)

184- how did you categorize the overall practice. Mention it in the methodology part and operational definition.

197- why did you use AOR for non- significant variables?

7. PLOS authors have the option to publish the peer review history of their article (what does this mean?). If published, this will include your full peer review and any attached files.

Reviewer #1: **Yes: **Shital Bhandary

Reviewer #2: **Yes: **Prem Prasad Panta

---

## [Author Response · Author response to Decision Letter 1]

22 Oct 2023

First of all, thank you all, for your nice comments and suggestions, many thanks for your genuine support and many help. We tried to correct the comments as the following: 

Reviewer #1

1. Line 115-16: The normality test was corrected with the Kolmogorov-Smirnov test 

2. Line 117-18: A higher p-value cut-off of 0.25 was used to transfer independent variables into the multivariable analysis, so as not to miss important variables in the final model (Reference 34) and we removed them from the result part. 

3. Line 117-19: Concerning sampling weight, the town has two kebeles (strata) and we did proportional allocations for each kebele (Kebele one had a proportional allocation of 304 mothers, and Kebele two had a proportional allocation of 116 mothers) (line 86 -87). In addition, in our study, the respondents or interviewers are only mothers among households. 

4. Line 119: Multiple regression analysis was changed with a multivariate logistic regression model

5. Line 123-24: VIF values were reported separately in Table 7 for each variable. The variance inflation factor greater than five was used as an indicator of multicollinearity (Reference 35)

Reviewer #2

1. Line 115: Corrected like this, the normality of the age variable was checked with the Kolmogorov-Smirnov Test (P<0.01) and the data was not normally distributed. 

2. Line 23: The sentence was corrected according to the given comments, the overall good practices of mothers towards the prevention of indoor air pollution was 188 (45.0%).

3. Line 80: Corrected with, Z value at 95% confidence interval =1.96

4. Line 87/88: The sampling interval was 19 (the total population of 8028 divided by a sample size of 420), then, the selected households were contacted for data collection. Since we had a sampling frame (lists) we did it easily on the paper, and then we identified and contacted for data collection.

5. Line 121: The value of the Hosmer and Lemeshow test was (P-value = 0.44) 

6. Line 123: The value of VIF corrected like this, the variance inflation factor greater than five was used as an indicator of multicollinearity. All the covariate values of VIF were less than one point five (VIF-value < 1.5), which showed the absence of collinearity among covariates. VIF values were reported separately in Table 7 for each variable. 

7. Line 142: Yes, in reality, the knowledge may vary with a huge range of age intervals, however, didn’t show significance with the outcome variable, which may be because of data quality

8. Line 184: Indoor air pollution prevention practices were categorized into good practice or poor practice by considering the percentage of mothers who correctly answered or fulfilled the given six questionnaires such as the use of a separate kitchen, opening a window while cooking, using of a modern stove, outside complete combustion of charcoal, separating under five children while cooking, and use of clean energy sources (Line 110-113).

9. Line 197: AOR for non-significant variables used to show more clarity, even though, some variables are transferred to the final model, didn’t indicate significance, therefore, the values are indicated for clarity. 

With all the best regards 

Worku Dugassa Girsha 

Thanks again for your genuine support!

---

## [Decision Letter · Decision Letter 2]

14 Nov 2023

PONE-D-23-21169R2Indoor air pollution prevention practices and associated factors among household mothers in Olenchiti town, Oromia, EthiopiaPLOS ONE

Dear Dr. Girsha,

Thank you for submitting your manuscript to PLOS ONE. After careful consideration, we feel that it has merit but does not fully meet PLOS ONE’s publication criteria as it currently stands. Therefore, we invite you to submit a revised version of the manuscript that addresses the points raised during the review process.

We look forward to receiving your revised manuscript.

Kind regards,

Srijan Lal Shrestha, Ph.D.

Academic Editor

PLOS ONE

Journal Requirements:

Reviewers' comments:

Reviewer's Responses to Questions

**Comments to the Author**

1. If the authors have adequately addressed your comments raised in a previous round of review and you feel that this manuscript is now acceptable for publication, you may indicate that here to bypass the “Comments to the Author” section, enter your conflict of interest statement in the “Confidential to Editor” section, and submit your "Accept" recommendation.

Reviewer #1: All comments have been addressed

2. Is the manuscript technically sound, and do the data support the conclusions?

Reviewer #1: Partly

3. Has the statistical analysis been performed appropriately and rigorously? 

Reviewer #1: No

4. Have the authors made all data underlying the findings in their manuscript fully available?

Reviewer #1: Yes

5. Is the manuscript presented in an intelligible fashion and written in standard English?

Reviewer #1: Yes

6. Review Comments to the Author

Reviewer #1: Line 84 - 91: This clearly requires the use of sampling weight so kindly calculate it in the excel and append it in the SPSS to do all the analysis using the sampling weight. Proportional allocation is ok but the way you have selected samples, replaced the household during field work and used lottery method to select mothers from households with more than one mother requires you to adjust the analysis using this design weight. It MUST be done! It does not matter how you did it (paper or electronically), it must be addressed now in the manuscript. Get help from a statistician who have done/worked with surveys in the past.

Line 125-126: The VIF cut-off of 5 is not correct for the logistic regression so use a reliable source and correct it. You are suggested to read "Applied Logistic Regression" book for the same.

Line 126-128: You must not describe your result here, do it where it must be done.

Table 7: It is not clear how you assessed the p-value < 0.25 for categorical variables like age categories, educational status, Occupation, Household monthly income as the overall p-value based on Wald test is not reported. This must be addressed.

7. PLOS authors have the option to publish the peer review history of their article (what does this mean?). If published, this will include your full peer review and any attached files.

Reviewer #1: **Yes: **Shital Bhandary

---

## [Author Response · Author response to Decision Letter 2]

24 Nov 2023

Responses to reviewer one 

1. Comment in lines 84–91, concerning sampling weight, different references indicated that sampling weights are the number of individuals in the population each respondent in the sample is representing. Accordingly, in our study, the respondents, or interviewers, are only mothers in households. However, only three households had no the required mother, and the next household was substituted. In addition, only two households had mothers with equal family representatives, selected by the lottery method. The detailed work was done in lines 84–91.

2. Comment in lines 125–126: Concerning the Variance Inflation Factor (VIF) cut-off point for logistic regression analysis, yes, we have seen different book references (evidences). Most research papers consider a VIF > 10 as an indicator of multicollinearity, but some choose a more conservative threshold of 5 or even 2.5. VIF > 5 is a cause for concern, and VIF > 10 indicates a serious collinearity problem. So that, we chose the more average conservative greater than five. The following references are attached:

a. Vittinghoff E. Regression Methods in Biostatistics: Linear, Logistic, Survival, and Repeated Measures Models. Springer; 2005.(Book)

b. ames G, Witten D, Hastie T, Tibshirani R. An Introduction to Statistical Learning: With Applications in R. 1st ed. 2013, Corr. 7th printing 2017 edition. (Book)

c. Menard S. Applied Logistic Regression Analysis. 2nd edition. SAGE Publications, Inc; 2001. (Book)

d. Johnston R, Jones K, Manley D. Confounding and collinearity in regression analysis: a cautionary tale and an alternative procedure, illustrated by studies of British voting behaviour. Qual Quant. 2018;52(4):1957-1976. doi:10.1007/s11135-017-0584-6, 2018 (Research paper)

3. Comment in line 126-128: Thanks, The variance inflation factors (VIF) result was taken under the regression result part (Lines 91–93).

4. Comment on Table 7. Thank you very much. We completely revised the regression table, and the overall p-value based on the Wald test to transfer candidates to the final model was revised according to your comments. Upon the revised value, the overall p-value for the age category (P<0.42) was greater than P<0.25, therefore the age category was omitted from the final model. Then the AOR with the P-value for each variable in the final model was rechecked and corrected. Based on revision, the educational level of diploma and above was not statistically significant (p<0.06). Furthermore, the result, discussion, conclusion, and recommendations were edited accordingly. Thank you once again for your professional input.

With all the best regards 

Worku Dugassa Girsha 

Thanks again for your support!

---

## [Decision Letter · Decision Letter 3]

18 Dec 2023

Indoor air pollution prevention practices and associated factors among household mothers in Olenchiti town, Oromia, Ethiopia

PONE-D-23-21169R3

Dear Dr. Girsha,

We’re pleased to inform you that your manuscript has been judged scientifically suitable for publication and will be formally accepted for publication once it meets all outstanding technical requirements.

Kind regards,

Srijan Lal Shrestha, Ph.D.

Academic Editor

PLOS ONE

Additional Editor Comments (optional):

Reviewers' comments:

Reviewer's Responses to Questions

**Comments to the Author**

1. If the authors have adequately addressed your comments raised in a previous round of review and you feel that this manuscript is now acceptable for publication, you may indicate that here to bypass the “Comments to the Author” section, enter your conflict of interest statement in the “Confidential to Editor” section, and submit your "Accept" recommendation.

Reviewer #1: All comments have been addressed

2. Is the manuscript technically sound, and do the data support the conclusions?

Reviewer #1: Yes

3. Has the statistical analysis been performed appropriately and rigorously? 

Reviewer #1: No

4. Have the authors made all data underlying the findings in their manuscript fully available?

Reviewer #1: Yes

5. Is the manuscript presented in an intelligible fashion and written in standard English?

Reviewer #1: Yes

6. Review Comments to the Author

Reviewer #1: The manuscript addressed all the comments but did not address the comments provided for the sampling weight.

7. PLOS authors have the option to publish the peer review history of their article (what does this mean?). If published, this will include your full peer review and any attached files.

Reviewer #1: **Yes: **Shital Bhandary

---

## [Editor Report · Acceptance letter]

10 Jan 2024

PONE-D-23-21169R3 

PLOS ONE

Dear Dr. Girsha, 

I'm pleased to inform you that your manuscript has been deemed suitable for publication in PLOS ONE. Congratulations! Your manuscript is now being handed over to our production team.

Kind regards, 

on behalf of

Dr. Srijan Lal Shrestha 

Academic Editor

PLOS ONE